# Ergonomic Assessment of a Lower-Limb Exoskeleton through Electromyography and Anybody Modeling System

**DOI:** 10.3390/ijerph19138088

**Published:** 2022-07-01

**Authors:** Yong-Ku Kong, Kyeong-Hee Choi, Min-Uk Cho, Seoung-Yoen Kim, Min-Jung Kim, Jin-Woo Shim, Sang-Soo Park, Kyung-Ran Kim, Min-Tae Seo, Hye-Seon Chae, Hyun-Ho Shim

**Affiliations:** 1Department of Industrial Engineering, Sungkyunkwan University, Suwon 16419, Korea; ykong@skku.edu (Y.-K.K.); ckhee@skku.edu (K.-H.C.); crayonmu@skku.edu (M.-U.C.); kimsy9035@skku.edu (S.-Y.K.); xlsk1013@skku.edu (M.-J.K.); sorksnrn33@skku.edu (J.-W.S.); qkrtkdtn53@skku.edu (S.-S.P.); 2National Institute of Agricultural Sciences, Rural Development Administration, Jeonju-si 54875, Korea; kimgr@korea.kr (K.-R.K.); mtseo85@korea.kr (M.-T.S.); hyeseon@korea.kr (H.-S.C.)

**Keywords:** lower-limb exoskeleton, CEX, AnyBody Modeling System, working height, drilling task, work-related musculoskeletal disorders

## Abstract

The aim of this study was to determine the muscle load reduction of the upper extremities and lower extremities associated with wearing an exoskeleton, based on analyses of muscle activity (electromyography: EMG) and the AnyBody Modeling System (AMS). Twenty healthy males in their twenties participated in this study, performing bolting tasks at two working heights (60 and 85 cm). The muscle activities of the upper trapezius (UT), middle deltoid (MD), triceps brachii (TB), biceps brachii (BB), erector spinae (ES), biceps femoris (BF), rectus femoris (RF), and tibialis anterior (TA) were measured by EMG and estimated by AMS, respectively. When working at the 60 cm height with the exoskeleton, the lower extremity muscle (BF, RF, TA) activities of EMG and AMS decreased. When working at the 85 cm height, the lower extremity muscle activity of EMG decreased except for TA, and those of AMS decreased except for RF. The muscle activities analyzed by the two methods showed similar patterns, in that wearing the exoskeleton reduced loads of the lower extremity muscles. Therefore, wearing an exoskeleton can be recommended to prevent an injury. As the results of the two methods show a similar tendency, the AMS can be used.

## 1. Introduction

Although recent developments have led to the automation of many processes in industrial sites, a wide variety of tasks that are difficult to replace by automation are still being carried out manually by workers. In industrial sites, workers are exposed to risk factors constantly, such as repetitive motion and awkward posture, which are the major causes of work-related musculoskeletal disorders (WMSDs) [1,2]. WMSDs cause pain in various body parts and occur frequently in automobile manufacturing line workers [3,4,5]. One study found that about 87.3% of automobile manufacturing workers experienced pain in at least one body part [6]; in particular, there were high rates of knee pain caused by kneeling and squatting postures [7,8]. Recently, the exoskeleton, a wearable device that supports the worker’s movement and posture, has emerged as an alternative to alleviate loads imposed by awkward postures, such as kneeling and squatting postures when working inside cars [9,10]. An exoskeleton is attached to the body to augment the human’s abilities or support the body. Industrial wearable robots are divided into various types, depending on how they operate and support body parts [9]. Exoskeletons can be classified into active and passive exoskeletons depending on how they work. Active exoskeletons use power (e.g., hydraulic or pneumatic pressure) to support the body, whereas passive exoskeletons use structure to support the body [9,11]. There are three main types of exoskeletons depending on the supported body parts: an upper-limb exoskeleton prevents excessive shoulder flexion in over-head working positions, a lower-limb exoskeleton prevents knee and trunk flexion in kneeling and bending working positions, and a lumbar support exoskeleton distributes lumbar loads throughout the body [12,13].

Many electromyography (EMG) studies have shown a reduction of muscle loads associated with wearing a passive exoskeleton, which is practical to use in the industrial and academic fields [14,15,16]. Kim et al. [14] showed that wearing an upper-limb exoskeleton decreases the load of shoulder muscles (middle deltoid and anterior deltoid). Pillai et al. [15] showed that when performing bolting tasks at the height of the hip and knee, wearing a lower-limb exoskeleton decreases the EMG amplitude of the lower-limb muscles (rectus femoris and tibialis anterior) and lumbar muscle (erector spinae). Yan et al. [16] showed that wearing the lower-limb exoskeleton when performing an assembly task from a squatting posture decreases the average muscle amplitude of the lower-limb muscles (rectus femoris, biceps femoris, vastus lateralis, and vastus medialis) by 71.5%.

In addition to evaluating the physical loads by EMG, many studies have used various biomechanical analysis programs that have been developed recently [17,18,19,20]. Panariello et al. [17] executed a risk analysis and comparative analysis with the Rapid Upper Limb Assessment (RULA) through body angle measurement using OpenSim during leveraging and drilling tasks. Using the Anybody Modeling System (AMS; Anybody Technology, Aalborg, Denmark), which has been used in many recent ergonomic studies, Zhou and Wiggerman [18] measured the load of a lumbar joint to show that when changing the posture of a patient lying on a bed, the load of the lumbar joint is proportional to the weight of the patient. Kong et al. [19] used the Three-Dimensional Statistical Strength Prediction Program (3D-SSPP) to evaluate the physical load when performing agricultural work, according to the weight and posture. Smith et al. [20] measured loads of the shoulder and elbow by analyzing the forces and movements associated with using cane-type medical aids.

In particular, the AMS, which is an inverse dynamic biomechanical analysis program recently used in the field of ergonomics for simulation of exoskeleton use, can calculate more than 500 loads occurring in skeletal muscles and joints based on acceleration of the human body and gravity [21]. EMG has the disadvantage of measuring only the muscles close to the skin surface, whereas the AMS software can analyze both the skin surface muscles and the deep muscles. In addition, unlike EMG measurements that require normalization of %MVC for each muscle group, AMS performs normalization by deriving maximum muscle activity values for each muscle group through the AnyBody Management Model Repository (AMMR), a dummy template provided by the program based on anthropometric information [22].

The primary purpose of this study is to evaluate the reduction of physical muscle loads during bolting tasks associated with wearing the lower-limb exoskeleton based on the experimental (EMG analysis) and simulated (AMS analysis) assessments. This study also evaluates the possibility of substituting a simulation analysis for the EMG analysis.

## 2. Materials and Methods

### 2.1. Participants

This study was conducted on 20 healthy men who had not experienced recent musculoskeletal pain or disorders. All participants were informed about the content and purpose of this study. Table 1 shows the anthropometric data of the participants. This research was approved by the Sungkyunkwan University IRB (approval#: SKKU 2021-01-007). 

### 2.2. Experimental Task

The bolting task, which is the most frequently performed task in the automobile assembly line, was performed at two working heights (knee flexion posture: 60 cm, back flexion posture: 85 cm) to verify the effectiveness of wearing the exoskeleton [14,15], as shown in Figure 1. The two working heights were selected based on data corresponding to the 50%-ile of men aged 20–59 years by referring to the 7th human body dimension data survey conducted by Size Korea [23]. The knee flexion posture (KF, 60 cm) was performed at 60 cm height, about 15 cm higher than knee height (45 cm). The back flexion posture (BF, 85 cm) was based on hip height (84.8 cm).

### 2.3. Apparatus

This study used a passive lower-limb exoskeleton called CEX developed by the Hyundai Motor Group (Seoul, Korea), which can be adjusted to seating angles of 55, 70, and 85°. Before the experiment, each participant had sufficient adaptation time to conduct tasks stably while wearing the CEX. Most of the participants set the angles to 55° for the knee flexion posture and 70° for the back flexion posture, as shown in Figure 1.

### 2.4. EMG

The TeleMyo 2400 DTS System (Noraxon, AZ, USA) was used to measure loads on muscles with and without the lower-limb exoskeleton CEX. Electrodes and sensors were attached to 16 muscle locations, on the left and right sides of 8 muscles, as measured in previous exoskeleton studies [15,24,25,26]. The upper-limb muscles of the middle deltoid (MD), biceps brachii (BB), triceps brachii (TB), and upper trapezius (UT); lumbar muscles of the erector spinae (ES); and rectus femoris (RF), tibialis anterior (TA), and biceps femoris (BF) were selected for monitoring, as illustrated in Figure 2. The electrode attachment position of each muscle was selected based on the guidelines of the surface EMG for the non-invasive assessment of muscles (SENIAM) [27]. Data were collected at 1500 Hz, and band-pass filtering at 20–400 Hz was performed for EMG analysis. Then, smoothing was performed by root-mean-square (100 ms). Since the intensity of the signal was different for each subject and muscle, the EMG was normalized by %MVC, a commonly used method in previous studies (Equation (1)) [28]:% MVC = [(EMG_task_ − EMG_resting_)/(EMG_max_ − EMG_resting_)] × 100 (%)(1)

### 2.5. The AnyBody Modeling System

The AMS software, which analyzes the loads on muscles and joints, is classified into three parts (Code, Chart, and Model), as illustrated in Figure 3. 

The Code integrates anthropometric information, motion data, the magnitude of external forces, and information on objects that combine and interact with the body. Motion data collected using the motion capture IMU sensors can be identified in the Code part of the software. This modification of the human model, input external force, and combined segment and body can also be performed through additional code modification input. The Chart shows time-series data, such as muscle activity and biomechanical data. The Model shows a visualized human model that provides various angles of body segments. The Model also can play or record motions.

Human motion data for AMS input data were recorded using an MVN Awinda (Xsens Technology B.V., Enschede, The Netherlands) Inertial Measurement Unit (IMU). Studies have shown that MVN Awinda has immunity to magnetic fields and produces fewer errors than other systems with IMU sensors [29]. Seventeen sensors were attached based on the Xsens guidelines, and data were collected at a 60 Hz sampling rate (Figure 4, left).

In addition, an ergoFET digital force gauge (Hoggan Scientific, Salt Lake City, UT, USA), a push–pull gauge, was installed on the wall to collect input force data generated at the end of the drill during the bolting task. The drilling force was measured two times for 2 s, and the average value was used in this study (Figure 4, right).

Since one of the purposes of this study was to measure the reduction of the muscle load associated with wearing the exoskeleton, it was necessary to combine the drill and lower-limb exoskeleton with the human body model.

Models of the drill and exoskeleton were designed using 3D computer-aided design (CAD) modeling, and conversion was performed by a plug-in, which was supported by AMS. The lower-limb exoskeleton consists of three parts (upper leg, lower leg, and a leg joint), and both sides were designed symmetrically. To implement mechanical movement, constraints were added the same way as the actual operation mechanism. Due to the coupling parts of the upper leg, lower leg, and leg joint rotating the hinge only, the movement was limited, except for the rotational motion in the corresponding direction. One end of the leg joint inserted into the groove of the upper leg only moved back and forth along the groove; a simple driver was used to limit movement, except for translational motion in the corresponding direction. Finally, the seating angle between the upper and the lower legs could be adjusted in the range 55–85°, the same as the CEX exoskeleton (Figure 5).

The drill was coupled to a glove node located in the hand of the human model through a standard point, and the external drilling force measured through a push–pull gauge was added to a tip node. The exoskeleton was located 3 cm away from the tuber node of the pelvis to combine a large part of the upper leg and the hip. The muscle activities of the eight muscles that were monitored with the EMG were calculated using AMS to estimate the muscle loads with and without the lower-limb exoskeleton (Figure 6).

### 2.6. Experimental Procedure

In this study, EMG data and motion capture data were collected to evaluate the reduction of muscle load associated with wearing the exoskeleton when performing the bolting task. Before the experiment, participants were informed of the purpose of the experiment and provided with sufficient time to practice wearing the exoskeleton. The anthropometric data of each participant were measured for motion capture (height, weight, shoulder height/width, elbow span, wrist span, arm span, hip height/width, knee height, and ankle height).

EMG electrodes were attached to the specified 16 locations (8 muscles, left/right) for EMG data measurement. After measuring the resting EMG and MVC, the participants performed four tasks (w/o Exo and w/ Exo at two working heights) for 10 min each. All tasks were randomized for each participant, and each task was repeated ten times with a 40 s bolting task and a 20 s break time (Figure 7). Finally, Xsens IMU sensors were attached to 17 body parts, and calibration was performed to collect motion data.

### 2.7. Statistical Analysis

For analysis of the results, exoskeleton wearing, working height, and muscle type were set as independent variables. Muscle activity (measured muscle activity from the EMG system and estimated muscle activity from AMS simulation) was set as a dependent variable. Muscle activity data used the average value of the left and right sides. ANOVA and paired *t*-test were performed to verify the difference between the dependent variables according to each independent variable, with the *p*-value at 0.05. Tukey’s test was then performed on significant variables. SPSS 25 (IBM, Armonk, NY, USA) [30] was used for all statistical analyses in this study.

## 3. Results

### 3.1. Experimental Assessment (EMG)—Measured Muscle Activity

As a result of the muscle activities measured by EMG, the main effects of wearing the exoskeleton, working height, and muscle type were statistically significant (all *p* < 0.05). Figure 8, left shows that when wearing the exoskeleton, the muscle activity (6.06% MVC) was 36.3% lower than when not wearing it (9.51% MVC). Figure 8, middle shows that when working at 85 cm (back flexion posture; BF_85), the muscle activity (7.27% MVC) was 12.3% lower than that (8.29% MVC) at the working height of 60 cm (knee flexion posture; KF_60). Figure 8, right shows that in terms of muscle type, UT, ES, and BF showed the highest muscle activity, while TB, MD, and RF showed the lowest muscle activity. 

Figure 9 shows that the interaction effect of wearing the exoskeleton and working height was also statistically significant (*p* < 0.05). In the knee flexion posture (KF_60), the average muscle activity decreased by 47.8% when wearing the exoskeleton (w/o Exo: 10.89% MVC vs. w/ Exo: 5.69% MVC). In the back flexion posture (BF_85), the muscle activity decreased by 20.9% when wearing the exoskeleton (w/o Exo: 8.12% MVC vs. w/ Exo: 6.42% MVC). Generally, the results of EMG analyses showed that when working at the lower height (i.e., KF_60), the effect of wearing the exoskeleton tended to be greater than when working at the height of 85 cm (BF 85).

Figure 10 shows that the interaction effect of wearing the exoskeleton and muscle type was also statistically significant (*p* < 0.05). Overall, all the muscle activities except BB showed a decreasing in muscle load when wearing the lower-limb exoskeleton. The muscle activities of the lower extremity (BF, RF, and TA) without exoskeleton were 17.33% MVC, 7.71% MVC, and 9.66% MVC, respectively, whereas the muscle activities of those muscles with exoskeleton were 5.22% MVC, 1.43% MVC, and 3.88% MVC, respectively, indicating significant decreases in muscle loads (69.9%, 81.5%, and 59.8%, respectively). On the other hand, the muscle activity of BB in the upper extremity increased by 33.1% (w/o exoskeleton: 6.73% MVC, w/ exoskeleton: 8.96% MVC).

Figure 11 shows that at the knee flexion posture (KF_60), the muscle activities of the upper extremity except for TB (UT, MD, and BB) and lumbar muscle (ES) tended to decrease when wearing the exoskeleton, while at the back flexion posture (BF_85), these muscle activities increased when wearing the exoskeleton.

Figure 12 shows that the muscle activities of the lower extremity (BF, RF, and TA) tended to decrease when wearing the exoskeleton, regardless of the working height. For example, at the back flexion posture (BF_85), the muscle activity of BF decreased by 82.38% when wearing the exoskeleton, whereas at the knee flexion posture (KF_60), the muscle activities of RF and TA decreased by 89.46% and 70.71%, respectively, when wearing the exoskeleton (*p* < 0.05).

### 3.2. Simulation Assessment (AnyBody Software)—Estimated Muscle Activity

As a result of the muscle activities estimated by AMS, the main effects of wearing the exoskeleton, working height, and muscle type were found to be statistically significant (all *p* < 0.05). Similar to the results of the experimental EMG assessment (Figure 8), when wearing the exoskeleton, the estimated muscle activity (5.80% MVC) was 17.5% lower than when not wearing it (7.03% MVC). When working at 85 cm, the muscle activity (5.90% MVC) was 14.9% lower than that (6.93% MVC) at the working height of 60 cm (Figure 13, left and middle).

In the case of the muscle type, Figure 13, right demonstrates that the lumbar muscle (ES) showed the highest muscle activity, followed by MD and BB, whereas TB and lower extremity muscles showed lower muscle activities.

Figure 14 shows that the interaction effect of wearing the exoskeleton and working height was also statistically significant (*p* < 0.05). In the knee flexion posture (KF_60), the muscle activity decreased by 23.9% when wearing the exoskeleton (w/o Exo: 7.87% MVC, w/ Exo: 5.99% MVC); while in the back flexion posture (BF_85), the muscle activity decreased by 9.21% when wearing the exoskeleton (w/o Exo: 6.19% MVC, w/ Exo: 5.62% MVC).

Figure 15 shows that the interaction effect of wearing the exoskeleton and muscle type was also statistically significant (*p* < 0.05). The muscle activity (18.35% MVC) of the lumbar muscle (ES) w/o Exo decreased to 15.01% MVC when wearing the lower-limb exoskeleton. The muscle activities of the lower extremity (BF, RF, TA) also showed decreasing trends of muscle loads (i.e., 4.0% MVC, 6.08% MVC, and 2.57% MVC w/o Exo vs. 0.23% MVC, 2.55% MVC, and 0.82% MVC w/ Exo, respectively). 

On the other hand, Figure 15 also shows that the muscle activities of the upper extremity (MD and BB) showed increasing trends of muscle loads. The MD and BB muscle activities w/o Exo were 10.57% MVC and 9.02% MVC, respectively, while the muscle activities w/ Exo were 12.50% MVC and 10.03% MVC, respectively. When working with the exoskeleton, the muscle activities increased by 18.3% and 11.2% for MD and BB muscle groups, respectively.

Figure 16 shows that when working at 60 cm, the estimated muscle activity of the upper extremity (MD) decreased by 13.0% (13.05% MVC vs. 11.36% MVC for w/o Exo vs. w/ Exo), whereas the muscle activity of the lumbar (ES) increased by 4.0% when wearing the exoskeleton (17.99% MVC vs. 18.71% MVC for w/o Exo vs. w/ Exo). 

On the other hand, this trend was reversed when working at 85 cm. When working at 85 cm, the muscle activity of the upper extremity (MD) increased by 68.8% (8.08% MVC vs. 13.64% MVC for w/o Exo vs. w/ Exo), whereas the muscle activity of the lumbar (ES) decreased by 39.6% when wearing the exoskeleton (18.70% MVC vs. 11.30% MVC for w/o Exo vs. w/ Exo).

Figure 17 shows that when wearing the exoskeleton, regardless of working heights, the estimated muscle activities of the lower extremities (BF and TA) decreased by 45.5–97.7%. On the other hand, the RF muscle showed a similar trend with MD, i.e., the estimated muscle activity of the RF decreased by 81.8% (12.08% MVC vs. 2.20% MVC for w/o Exo vs. w/ Exo) when working at 60 cm, whereas the estimated muscle activity of the RF increased from 0.08% MVC to 2.90% MVC for w/o Exo vs. w/ Exo when working at 85 cm.

## 4. Discussion

An exoskeleton is a kind of support that can assist the worker in reducing the worker’s muscle load and preventing excessive flexion or extension of the body in many tasks, such as bending, squatting, and lifting. Thus, exoskeletons have recently been developed, applied, and evaluated in various industrial fields. In this study, muscle activities were investigated using EMG and the AMS to evaluate muscle loads at different working heights involving back and knee flexion postures with and without use of a lower-limb passive exoskeleton. These back and knee flexion postures occur frequently in automobile assembly lines.

### 4.1. Comparison of the Effect of Lower-Limb Exoskeleton

Wearing the lower-limb exoskeleton decreased the muscle activities by 36.3% (EMG) and 17.5% (AMS) in this study. These findings are similar to the results of previous studies that investigated muscle activities associated with wearing a lower-limb exoskeleton. Previous studies showed that the average muscle activity decreased by 20.7–72.7% when wearing the exoskeleton [16,24,26,31]. 

Yan et al. [16] showed that the muscle activities of RF and BF decreased by 66.1% and 72.7%, respectively, when wearing a lower-limb passive exoskeleton during the squatting posture (i.e., 83.7% MVC and 85.6% MVC w/o Exo vs. 28.4% MVC and 23.3% MVC w/Exo). Huysamen et al. [24] showed that for 7.5 kg and 15.0 kg lifting tasks, the muscle activities of ES and BF decreased by 30.0% and 30.8%, respectively, (7.5 kg lifting task), and 27.5% and 21.0%, respectively, (15 kg lifting task) when wearing an exoskeleton (i.e., 40.6% MVC and 19.3% MVC w/o Exo vs. 28.4% MVC and 13.3% MVC w/ Exo for 7.5 kg lifting task; 53.7% MVC and 24.0% MVC w/o Exo vs. 39.0% MVC and 19.0% MVC w/ Exo for 15 kg lifting task). Bosch et al. [31] also showed that the muscle activities of ES and BF decreased 34.8% and 20.7%, respectively, during assembly work (i.e., 10.9% MVC and 11.7% MVC w/o Exo vs. 7.1% MVC and 9.3% MVC w/ Exo) and 35.8% and 24.4%, respectively, during upper-limb flexion posture (i.e., 10.3% MVC and 10.4% MVC w/o Exo vs. 6.6% MVC and 7.9% MVC w/ Exo) when wearing an exoskeleton. 

As expected from Figure 6 (left), the knee flexion posture (KF_60) had greater back and knee flexion angles than the back flexion posture (BF_85) without an exoskeleton. Thus, the average muscle activity (10.2% MVC) at the KF_60 working height was 25.9% higher than that (8.1% MVC) at the BF_85 working height in the EMG analysis. The UT, RF, and TA muscle activities (20.6% MVC, 13.3% MVC, and 16.7% MVC) of the KF_60 working height were higher than those (4.6% MVC, 2.1% MVC, and 2.5% MVC) of the BF_85 working height, whereas the BF muscle activities (7.1% MVC) of the KF_60 were significantly lower than that (27.5% MVC) of the BF_85 working height. The AMS analysis also showed similar results: the average muscle activity (7.9% MVC) at the KF_60 working height was 27.4% higher than that (6.2% MVC) at the BF_85 working height. The MD, RF, and TA muscle activities (13.1%, 12.1%, and 3.3% MVCs) of the KF_60 working height were higher than those (8.1%, 0.1%, and 1.9% MVCs) of the BF_85 working height, whereas the BF muscle activities (1.5% MVC) of the KF_60 working height were lower than that (6.5% MVC) of the BF_85 working height. 

The lower working postures without an exoskeleton had greater back and knee flexion angles with an extension of the front part of the leg, resulting in large muscle loads on RF and TA muscles. Also, the higher working postures had smaller knee angles with an extension of the back part of the leg, resulting in large muscle loads on the BF muscle in this study.

EMG and AMS analyses both indicated that the reductions of muscle loads when wearing the lower-limb exoskeleton were greater at the working height of 60 cm (KF_60) than at 85 cm (BF_80). The muscle activities decreased by 47.8% and 23.9% (EMG and AMS) at KF_60, as compared to 20.9% and 9.2% (EMG and AMS) at BF_85 (Figure 9 and Figure 14). These results are consistent with the findings of previous studies that provided guidelines for applicable exoskeleton heights when performing harvesting tasks. Kong et al. [26] compared the muscle activity data of the whole body when wearing a lower limb exoskeleton when performing harvest simulation at six working heights ranging from 40 to 140 cm. As a result of wearing the lower-limb exoskeleton at low working heights of 40, 60, and 80 cm, muscle activity significantly decreased by 36.0%, 31.3%, and 24.4%, whereas the muscle activities tended to increase when wearing the exoskeleton at higher working heights of 100, 120, and 140 cm. From these results, it seems that the lower the working height, the greater the effect of the lower-limb exoskeleton on the reduction of muscle load. Thus, the authors recommended wearing the lower-limb exoskeleton at working heights less than 80 cm.

In this study, the analyses of EMG and AMS showed a decreasing trend of muscle activities for the lumbar muscle (ES) and lower extremity muscles (RF, BF, and TA) and an increasing trend of the upper extremity (BB) associated with wearing the exoskeleton. Since the main purpose of the lower-limb exoskeleton is to reduce loads of the lower extremity muscles, these results are promising. As mentioned before, the activities of the lower extremity muscles decreased when wearing the lower-limb exoskeleton in the analyses of both EMG and AMS. In the case of EMG, the muscle activity of BF decreased by 82.4% (27.5% MVC w/o Exo vs. 4.9% MVC w/ Exo) at the working height of 85 cm (BF_85), while those of RF and TA decreased by 89.5% and 70.7% (13.3% MVC and 16.7% MVC w/o Exo vs. 1.4% MVC and 4.9% MVC), respectively, at the working height of 60 cm (KF_60). In the case of AMS, the muscle activities of BF, RF, and TA decreased by 78.2%, 81.8%, and 81.1% (1.5% MVC, 12.1% MVC, and 3.3% MVC w/o Exo vs. 0.3% MVC, 2.2% MVC, and 0.6% MVC w/Exo), respectively, at the working height of 60 cm (KF_60), while those of BF and TA decreased by 97.7% and 45.5% (6.5% MVC and 1.9% MVC w/o Exo vs. 0.2% MVC and 1.0% MVC), respectively, at the working height of 85 cm (BF_85). It was noted that wearing the exoskeleton at a low working height (60 cm) tended to decrease the muscle activities of RF and TA more, whereas wearing the exoskeleton at a high working height tended to decrease the muscle activity of BF more. 

### 4.2. Contribution

The muscle activities of eight muscle groups, including the upper-limb, back muscle, and lower-limb muscles, were analyzed by the AMS in the current study. Generally, the muscle activity of the AMS analysis tended to be lower than that of EMG, but the overall trends in muscle activities associated with wearing the exoskeleton were similar for the two methods. The AMS method has been used recently in ergonomic analysis due to the advantage of not requiring additional experimental devices and possible repeated performance to investigate trends of actual muscle activities (EMG). It can be seen that the tendency of muscle activity at a low working height (the knee-flexion posture) and a high working height (the back-flexion posture) according to wearing the exoskeleton was similar to that of EMG measurements. In particular, in the case of the lower leg muscles (BF, RF, and TA), both EMG and AMS results showed a significant decrease in muscle activity at the knee flexion posture when wearing the exoskeleton. Lee et al. [32] showed that the muscle activity of RF using EMG and AMS during the squatting posture had similar patterns. Alexander and Schwameder [33] also showed that the muscle activities of leg muscles (BF, RF, and TA) showed a high correlation coefficient between the results of EMG and AMS when walking along a slope (i.e., 0.82, 0.7, and 0.51 at 18° slope; 0.75, 0.67, and 0.7 at 12° slope). Thus, AMS might be an alternative method for analyzing muscle activity.

This study compared the analyses of muscle activities by EMG and the AMS to evaluate the muscle load reduction associated with wearing the exoskeleton when performing bolting tasks at two working heights, as frequently performed on assembly lines. This study suggests that the use of wearable lower-limb exoskeletons in the workplace could reduce the incidence of musculoskeletal disorders in the lower limbs. Additionally, this study presents an alternative to using the existing EMG measurement system. The possibility of using a new simulation analysis method to derive results similar to existing EMG methods has been investigated. 

### 4.3. Limitation and Future Work

The first limitation of this study is that only a static task was performed at fixed working heights for men in the laboratory environment. Thus, it is necessary to conduct additional experiments on dynamic tasks and different working environments with participants of various age and gender groups to evaluate the actual applicability. In addition, this study was conducted with only 20 participants. This sample size could cause statistical problems such as small statistical power. Therefore, it is necessary to include additional participants.

The second limitation is that this study did not consider the cognitive aspects. There have been many studies that show a relationship between physical loads and cognitive capacity. Dietrich and Sparling [34] showed when running on treadmills at 70–80% of maximum heart rate, performance of attention tasks such as the Wisconsin Card Sorting Task was degraded. Remaud et al. [35] also showed the relationship between physical loads and cognitive capacity. This study showed that when standing with both legs or a single leg, reaction time to an auditory stimulus was greater than when seating. Therefore, it may be valuable to study the relationship between physical loads measured by AMS and cognitive capacity.

## 5. Conclusions

The purpose of this study was to show the muscle load reduction according to wearing the lower-limb exoskeleton by using experimental assessment (EMG) and simulation assessment (AMS). The results showed that regardless of working heights, wearing the exoskeleton could reduce the overall muscle loads. However, this tendency was shown to be somewhat greater at a working height of 60 cm. Considering each body part, the loads on the lower extremity muscles tended to decrease with the addition of the exoskeleton at both working heights, while the upper extremity muscles tended to increase or remain constant at a working height of 85 cm. Therefore, considering the lower extremity muscles and the upper extremity muscles, it is recommended that workers wear an exoskeleton to prevent musculoskeletal disorders when working at a low height. In addition, since EMG and AMS analyses of muscle activity patterns showed similar trends in the lower-limb muscles (BF, RF, and TA), it seems possible to use AMS as an additional option for measuring and analyzing the muscle load.

## Figures and Tables

**Figure 1 ijerph-19-08088-f001:**
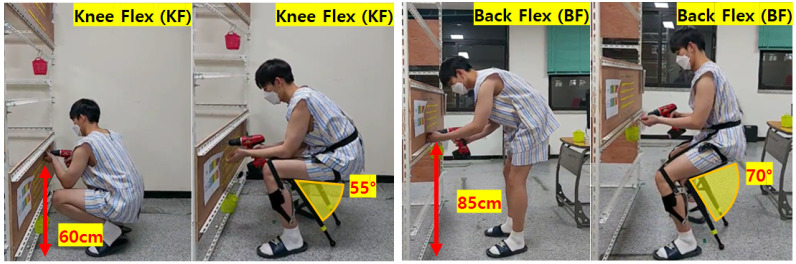
The two working heights used in the bolting task (w/ and w/o CEX).

**Figure 2 ijerph-19-08088-f002:**
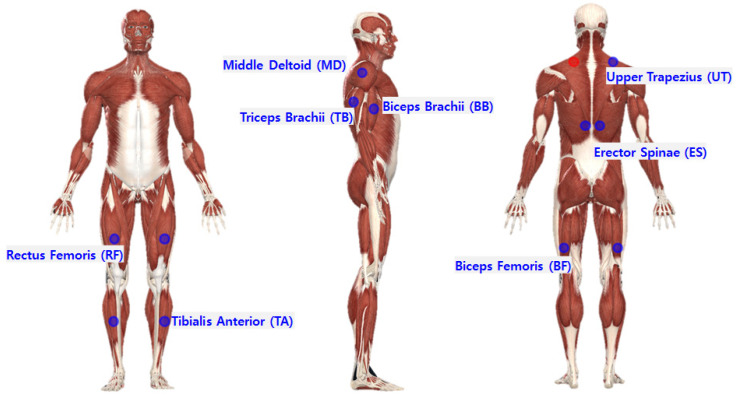
Locations of EMG sensor attachments.

**Figure 3 ijerph-19-08088-f003:**
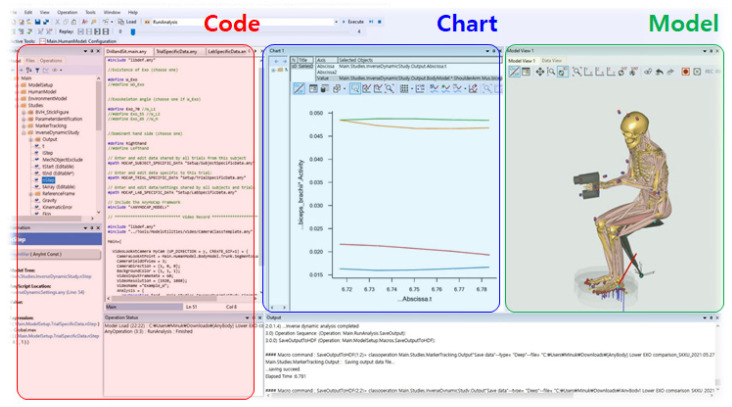
Structure of the AnyBody Modeling System.

**Figure 4 ijerph-19-08088-f004:**
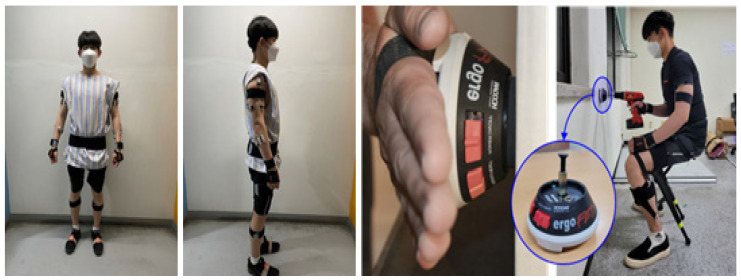
Motion capture system (MVN Awinda, **left**) and drilling force measurement system (ergo FET, **right**).

**Figure 5 ijerph-19-08088-f005:**
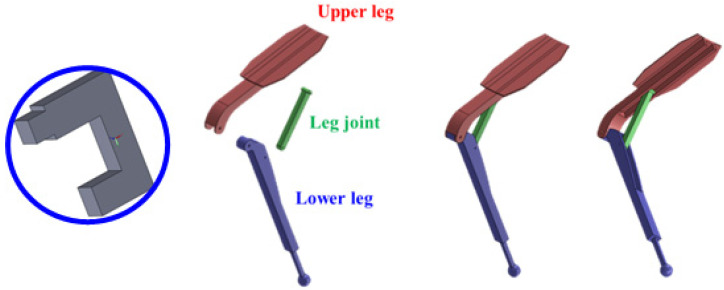
3D CAD models of drill (**left**), and lower-limb exoskeleton (**right**).

**Figure 6 ijerph-19-08088-f006:**
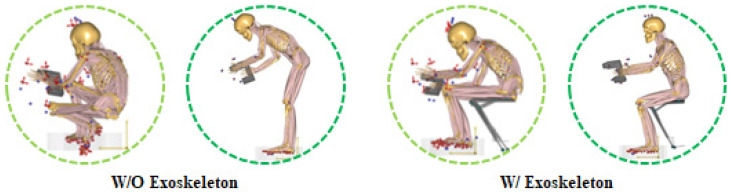
AMS models without (**left**) and with (**right**) exoskeleton during bolting tasks.

**Figure 7 ijerph-19-08088-f007:**
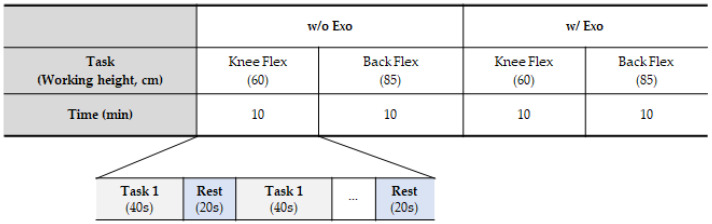
Drilling task procedure (all tasks were performed randomly for each subject).

**Figure 8 ijerph-19-08088-f008:**
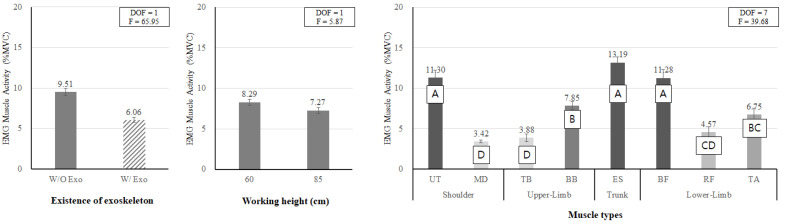
The main effects of wearing the exoskeleton (**left**), working height (**middle**), and muscle type (**right**) on EMG muscle activity. Different letters indicate significant statistical differences (*p* < 0.05, Tukey’s test).

**Figure 9 ijerph-19-08088-f009:**
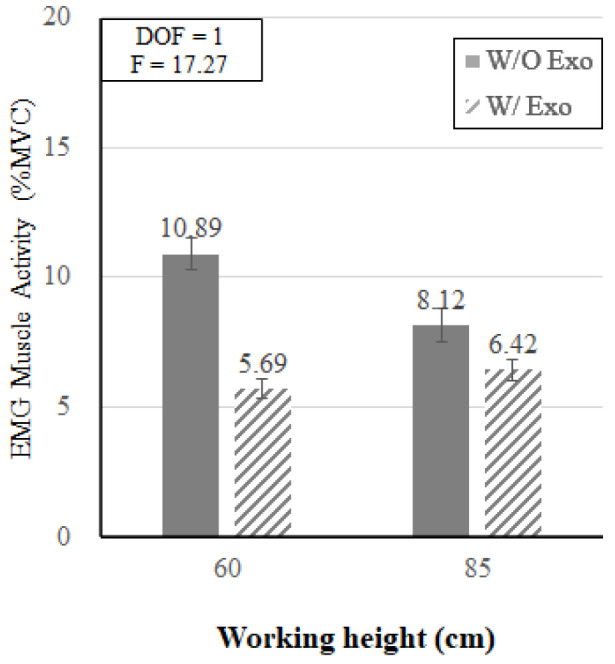
The interaction effect of wearing the exoskeleton and working height.

**Figure 10 ijerph-19-08088-f010:**
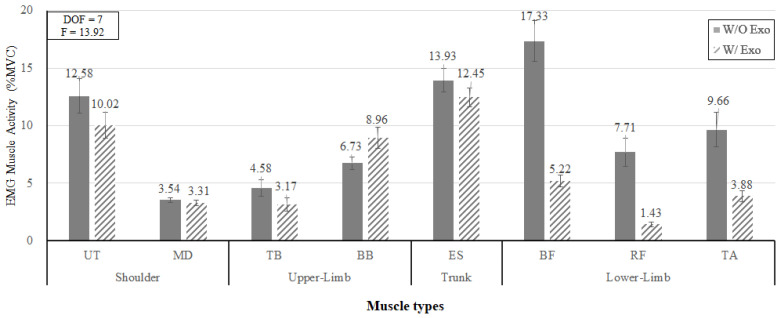
The interaction effect of wearing the exoskeleton and muscle type.

**Figure 11 ijerph-19-08088-f011:**
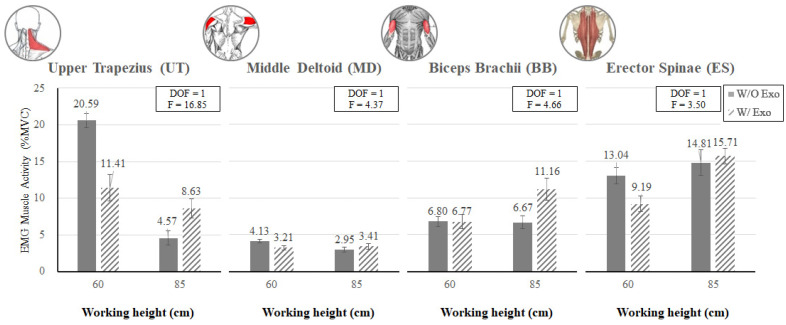
The interaction effects of wearing the exoskeleton, working height, and muscle type (UT, MD, BB, and ES) on experimental muscle activity.

**Figure 12 ijerph-19-08088-f012:**
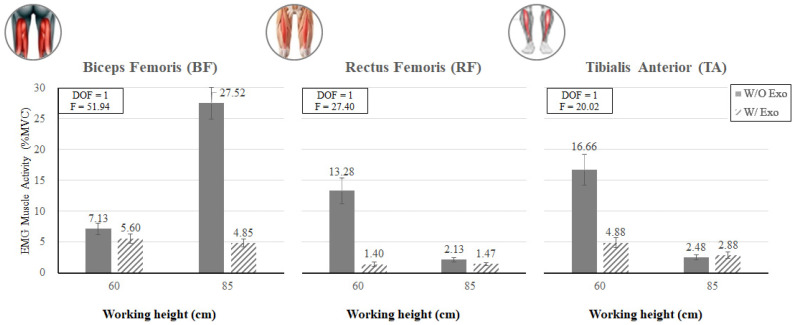
The interaction effects of wearing the exoskeleton, working height, and muscle type (BF, RF, and TA) on experimental muscle activity.

**Figure 13 ijerph-19-08088-f013:**
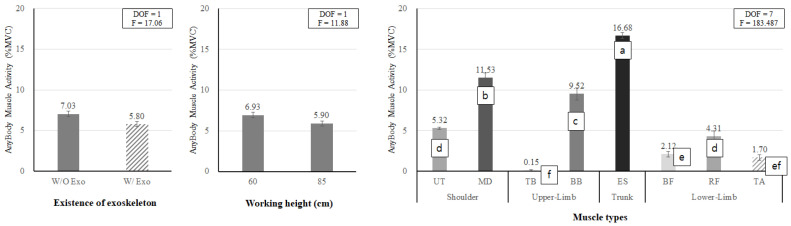
Main effects of wearing the exoskeleton (**left**), working height (**middle**), and muscle type (**right**) on simulated muscle activity. Different letters indicate significant statistical differences (*p* < 0.05, Tukey’s test).

**Figure 14 ijerph-19-08088-f014:**
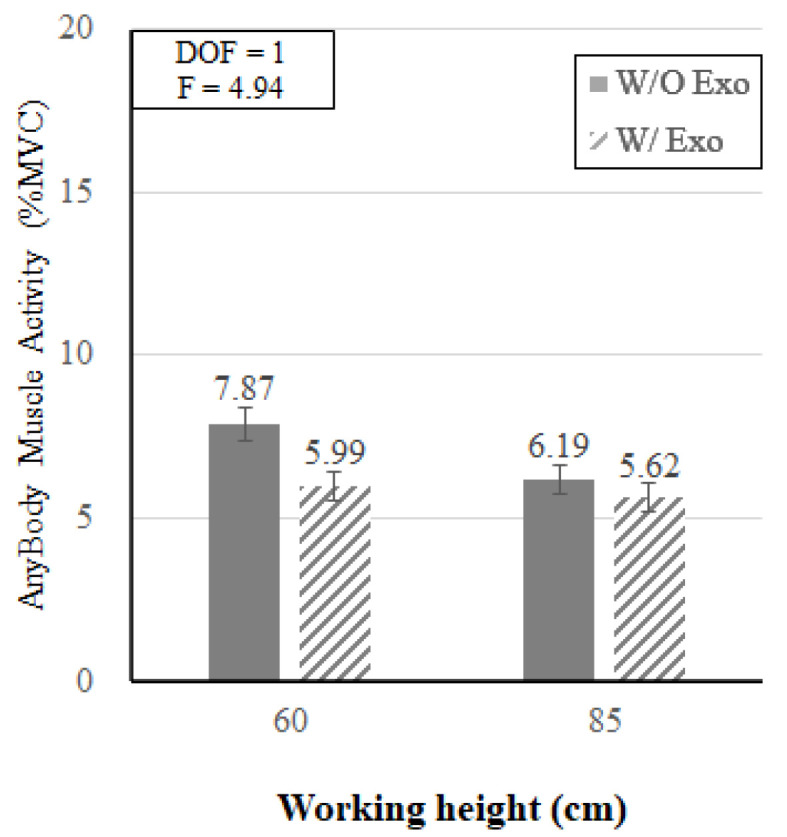
Interaction effect of wearing the exoskeleton and the working height on simulated muscle activity.

**Figure 15 ijerph-19-08088-f015:**
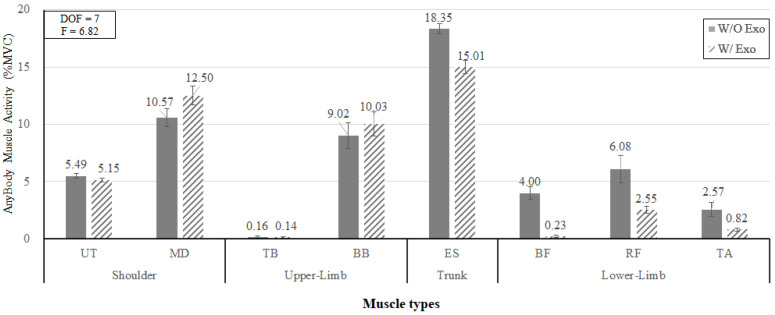
The interaction effect of wearing the exoskeleton and muscle type on the simulated muscle activity.

**Figure 16 ijerph-19-08088-f016:**
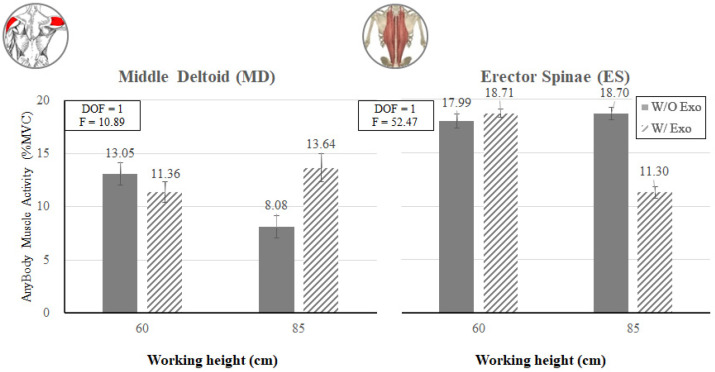
Interaction effects of wearing the exoskeleton, working height, and muscle type (MD and ES) on the simulated muscle activity.

**Figure 17 ijerph-19-08088-f017:**
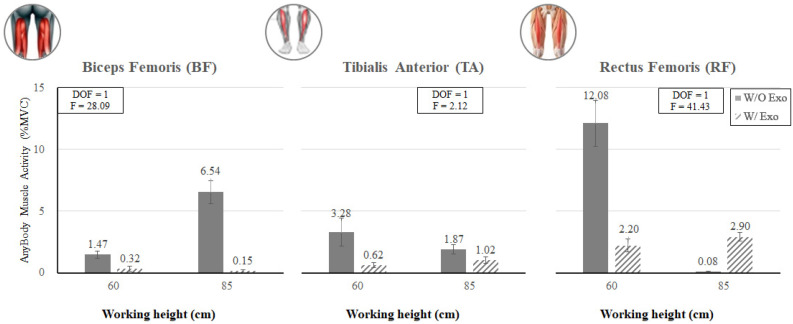
Interaction effects of wearing the exoskeleton, working height, and muscle type (BF, TA, and RF) on the simulated muscle activity.

**Table 1 ijerph-19-08088-t001:** Anthropometric data of the participants (mean ± standard deviation).

Age (years)	Height (cm)	Weight (kg)
24.8 ± 2.5	176.4 ± 3.8	78.8 ± 12.0

## Data Availability

Not applicable.

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
