# Peer review of "Ergonomic Assessment of a Lower-Limb Exoskeleton through Electromyography and Anybody Modeling System"

_ijerph, 2022, doi:10.3390/ijerph19138088_

Round 1
Reviewer 1 Report
I enjoyed reading the paper. The authors have presented an interesting article about an Ergonomic Assessment of a Lower-limb Exoskeleton through Electromyography and Anybody Modeling System.
Abstract, overview
The abstract is a concise description of the work. The introduction is well structured, and it covers all the concepts investigated in the methodological part. The previous work is well presented and integrated. I consider that this work brings added value in the field and the specific objectives of the manuscript are well related to the previous work developed in this domain.
Methodology
The research design used is appropriate in order to answer the research questions proposed by the authors. The methods are described properly. The results are clearly presented and are in relation to the concepts investigated.
Discussion and conclusions
The discussions are clear and concise. The conclusions are strongly related to the findings of the research work.
Format and style
All the format and style features were respected and are compliant with the requirements.
References
The format of the reference list fixes well to the specified format.
Plagiarism and any other ethical concerns about this study
I do not have any potential conflict of interest with regards to this paper.
Despite the good work done, there is still some room for improvement, as follows:
- How do you think your findings can be useful applicable to other engineering disciplines? I think some more literatures should be added. Several biological data process system is applied nowadays. It would be good to see the "effect of different web-based media" content on "human brain waves", as well as the additional applications of brainwave-based control. It would improve the quality of the publication to mention the relationship between a cognitive psychological attention test and the attention levels determined by a BCI systems such as in an examination and comparison of the EEG based attention test with CPT and TOVA. In addition to BCI systems, mentioning other important human-computer interaction eye movement tracking would also improve quality, as such systems can be used in the analysis of programming technologies such as LINQ and algorithms, thus enabling, for example, cognition load or source code, algorithm description tools readability testing like in measuring cognition load using eye-tracking parameters based on algorithm description tools, in clean and dirty code comprehension by eye-tracking based evaluation using GP3 eye tracker and in analyse the readability of LINQ Code using an eye-tracking-based evaluation.
Reviewer 2 Report
Dear Authors, I found this study an interesting endeavour to evaluate the reduction of physical muscle 85 loads during bolting tasks associated with wearing the lower-limb exoskeleton. I find it mostly well-written in terms of writing and statistical perspectives. I think it needs some extra work to arrive to publication quality. Regards. P.S.
[1] Writing:
1-1 Discussion Section: Its is written in imbalanced way that when reading it the readers loses the "road map" . Please re-organize it for smooth reading by sub-sectioning it. Something like this: 4.Discussion; 4.1. This Work, 4.2. Contributions and Limitations, 4.3. Future Work
1-2 Abbreviations: Please add the list of used abbreviations in the work in end right before reference section for easy readers referral. Example: Abbreviation: AMS: AnyBody Modeling System ; etc.
[2] Statistical:
2-1 Low Sample Size: The study has only 20 participants. This may cause low power problem. Please report power of each conducted test in the work.
2-2 SPSS citation: In line 221, it needs its own citation in the reference section: *IBM Corp. Released 2020. IBM SPSS Statistics for Windows, Version 25.0. Armonk, NY: IBM Corp
2-3 Limitations: Low sample size and lack of control for potential confounders (e.g., age, gender, race, etc.) are additional limitations of the work. Need to mention this in subsection "4.2. Contributions and Limitations"
Round 2
Reviewer 1 Report
The authors answered my questions more or less.But several biological data process / sensors or systems are applied nowadays.
It would be good to see the "effect of different web-based media" content on "human brain waves", as well as the additional applications of brainwave-based control. It would improve the quality of the publication to mention the relationship between a cognitive psychological attention test and the attention levels determined by a BCI systems such as in an examination and comparison of the EEG based attention test with CPT and TOVA. In addition to BCI systems, mentioning other important human-computer interaction eye movement tracking would also improve quality, as such systems can be used in the analysis of programming technologies such as LINQ and algorithms, thus enabling, for example, cognition load or source code, algorithm description tools readability testing like in measuring cognition load using eye-tracking parameters based on algorithm description tools, in clean and dirty code comprehension by eye-tracking based evaluation using GP3 eye tracker and in analyse the readability of LINQ Code using an eye-tracking-based evaluation.
Author Response
This manuscript is a study of the effects of using Exoskeletons on muscle activities based on EMG (muscle activity) and AMS (simulation).
However, the reviewer's comments are very different from our research goals and contents. He mentioned cognitive issues, including BCI, CPT, TOVA, EEC-based attention, and so on, which are way from our research topics.